# Left ventricular mass normalization in child and adolescent athletes must account for sex differences

Hubert Krysztofiak[1,2]*, Marcel Młyńczak[3], Łukasz A. Małek[4], Andrzej Folga[2], Wojciech Braksator[5]

1 Mossakowski Medical Research Centre, Polish Academy of Sciences, Warsaw, Poland, 2 National Centre for Sports Medicine, Warsaw, Poland, 3 Faculty of Mechatronics, Institute of Metrology and Biomedical Engineering, Warsaw University of Technology, Warsaw, Poland, 4 Department of Epidemiology, Cardiovascular Disease Prevention and Health Promotion, National Institute of Cardiology, Warsaw, Poland, 5 Department of Sports Cardiology and Noninvasive Cardiovascular Imaging, 2nd Medical Faculty, Medical University of Warsaw, Warsaw, Poland

* hkrysztofiak@imdik.pan.pl

**Data Availability Statement:** All relevant data are within the paper and its Supporting Information files.

## Abstract

### Background

To assess left ventricular hypertrophy, actual left ventricular mass (LVM) normalized for body size has to be compared to the LVM normative data. However, only some published normative echocardiographic data have been produced separately for girls and boys; numerous normative data for the pediatric population are not sex-specific. Thus, this study aimed to assess whether the LVM normative data should be developed separately for girls and boys practicing sports.

### Methods

Left ventricular mass was computed for 331 girls and 490 boys, 5–19 years old, based on echocardiography. The effect of sex on the relationship between LVM and body size was evaluated using a linear regression model. Seven sets of the LVM normative data were developed, using different methodologies, to test concordance between sex-specific and non-specific normative data. Every set consisted of normative data that was sex-specific and non-specific. Upon these normative data, for every study participant, seven pairs of LVM z-scores were calculated based on her/his actual LVM. Each pair consisted of z-scores computed based on sex-specific and non-specific normative data from the same set.

### Results

The regression lines fitted to the data points corresponding to LVM of boys had a higher slope than of girls, indicating that sex affects the relationship between LVM and body size. The mean differences between the paired LVM z-scores differed significantly from 0. The percentage of discordant indications, depending on the normalization method, ranged from 66.7% to 100% in girls and from 35.4% to 50% in boys. Application of the LVM normative

**Funding:** The authors received no specific funding for this work.

**Competing interests:** The authors have declared that no competing interests exist.

data that were not sex-specific made relative LVM underestimated in girls and overestimated in boys.

## Conclusion

The LVM normative data should be developed separately for girls and boys practicing sports. Application of normative data that are not sex-specific results in an underestimation of relative LVM in girls and overestimation in boys.

## Introduction

Echocardiography is recommended as a first-line diagnostic tool for cardiac size evaluation in both children and adults [1–3]. Direct measurements of left ventricular muscle and chamber dimensions and further computation of left ventricular mass (LVM), provide the basis to diagnose hypertrophy [2,4].

In general, the presence of left ventricular hypertrophy (LVH) is associated with increased risk for adverse cardiovascular (CV) outcomes [5,6]. In turn, in athletes, LVH is recognized as an adaptive and physiological feature related to exercise [7–9]. Although exercise is considered as a strong counter-measure against cardiovascular disease (CVD) [10–13], athletes and physically active people are not CV disease-immune population. Sometimes in athletes, LVH has to be differentiated when there is suspicion or presence of such clinical problems as hypertrophic cardiomyopathy (HCM), hypertension, or valvular insufficiency or stenosis [3,14,15].

Since cardiac size depends on body size, the normalization for body size is necessary for better left ventricular (LV) assessment; the absolute values of LVM or other LV measures have to be normalized upon body size variable [2,16]. Such normalization is especially needed for children and adolescents because of continuous body size changes during development and the significant differences in body size, even between children of the same age [1].

To diagnose or manage LVH, actual LVM normalized for body size has to be compared to properly developed normative data. In our recent studies, we analyzed a problem of the correct body size variable for cardiac LVM normalization [17] and the issue of an accurate and convenient method for cardiac size scaling [18]. In previous studies, we developed normative data for children and adolescents practicing sport and compared them to the normative data generated for the general pediatric population [19,20]. When we were working on the topic of LVM normalization for body size, a question arose: should LVM normative data, or generally normative data of the LV dimensions, be sex-specific or not?

A review of articles published on this issue shows that this question is still open. In most of the works related to the pediatric population, the developed echocardiographic normative data are not sex-specific. Sometimes, there is no information on why they are developed in this way [21,22]. Sometimes, the authors evaluated the influence of sex on the result of the measurement and found no significant effect [23–25]. Some other times, the authors found a significant statistical effect but recognized it as not meaningful from a clinical perspective [26]. On the other side, there are important works where sex-specific LVM normative data were presented because researchers noted significant differences in relative LVM between boys and girls [27–30].

An interesting issue is a search for a body size variable for which the effect of sex on the relationship with LVM is not significant. Such a variable, used as an explanatory one, would enable generating normative data for the pooled population, without division on sex. It is

considered that lean body mass has such properties and that allometrically adjusted height might give the same result [31,32].

Thus, should LVM normative data be sex-specific or not? To provide a substantive basis for answering this question, we designed a study to explore the effect of sex on the relationship between LVM and body size variables used in cardiac size scaling and evaluate the concordance between sex-specific and non-specific LVM normative data developed according to different methods. The study aimed to assess whether, for reliable evaluation of LVM in children and adolescents, left ventricular mass normative data should be developed separately for girls and boys.

## Materials and methods

### The study participants

It was a retrospective study conducted as a continuation of our previous works on normative echocardiographic data of LVM for youth athletes and the methodology of LVM normalization upon body size. The medical data used in this study were collected between 2013 and 2018. The study participants were children and adolescents practicing sport, engaged in regular athletic training at the local or national level. Starting with the most popular sports, they had practiced: soccer, swimming, basketball, handball, fencing, rowing, tennis, dancing, distance running, speed skating, cycling, sailing, and martial sports like karate, taekwondo, judo, and wrestling. Since these were child and adolescent athletes, predominantly amateur, their training was focused mainly on general physical performance, building the aerobic capacity, and motor skills. They were examined during periodic preparticipation physical evaluation. The study group consisted of 331 girls and 490 boys (821 children and adolescents in total). Both girls and boys were aged from 5 to 19 years. All the participants were white. Echocardiography was ordered because of innocent heart murmurs or suspicion of abnormal electrocardiographic findings. However, the athletes in whom echocardiography revealed significant acquired or congenital heart diseases, affecting heart size and hemodynamics, were not included in the study. Height and body mass were measured during the main examination.

### Echocardiography

Echocardiographic examinations were performed by experienced sonographers using a commercially available ultrasound scanner (Toshiba Aplio 400, Toshiba Medical Systems Europe, Zoetermeer, the Netherlands), according to recent guidelines. All measurements were taken in the 2-dimensional parasternal long-axis view (PLAX) at end-diastole and included the basic linear cardiac dimensions necessary for LVM computing: left ventricular internal dimension (LVIDd), interventricular septal thickness (IVSd), and posterior wall thickness (PWTd). All measurements were taken from the inner edge to inner edge and reported to within 1 mm. Left ventricular mass was computed according to the formula of Devereux et al. [4]:

$$LVM = 0.8\{1.04[(LVIDd + PWTd + IVSd)^3 - (LVIDd)^3]\} + 0.6$$

### Ethical considerations

The Ethics Committee of the Medical University of Warsaw approved the study procedure (approval AKBE/75/17). As the study was retrospective, and the data used were collected during routine medical monitoring, neither written nor verbal consent was required for this particular study. However, each subject, or the subject's parent or legal guardian, had signed the informed consent form for the routine medical monitoring, including a statement of agreement to the use of the results for scientific purposes.

## Evaluation of the relationship between LVM and body size in girls and boys

We started with an evaluation of the effect of sex on the relationship between LVM and body size. For each selected body size parameter, we compared two regression lines representing this relationship for girls and boys, respectively. To perform distinct analyzes, we selected four body size variables: body mass, height, body surface area (BSA), and computed lean body mass (cLBM). Body surface area was calculated according to the Haycock formula [33], and cLBM was calculated based on equations introduced by Foster et al. [34].

At first, we inspected graphical presentations of these relationships. Separate scatter plots of LVM against each body size variable were drawn. On every scatter plot, both girls' and boys' data points were presented, and specific regression lines were fitted, respectively. To compare $y$-intercepts and slopes of the two regression lines, we constructed a linear regression model introducing a dummy variable representing sex. We coded girls as 0, and boys as 1. In our model, LVM was a dependent variable, and a body size parameter, the dummy variable, and the product of the body size parameter and the dummy variable were independent variables. For combined groups of girls and boys, the model was expressed using the following equation:

$$y = \beta_0 + \beta_1 x + \beta_2 z + \beta_3 xz$$

where $y$ is LVM, $x$ is body size variable, and $z$ is the dummy variable. So, when $z = 0$, the regression is:

$$y = \beta_0 + \beta_1 x$$

and for $z = 1$, it will be:

$$y = (\beta_0 + \beta_2) + (\beta_3 + \beta_1)x$$

If the coefficient $\beta_2$ in the presented model is different from 0, it means that the $y$-intercepts of the sex-specific regression lines are different; there is a fixed difference in LVM between girls and boys, across the whole range of body size parameter's magnitude. If the coefficient $\beta_3$ (interaction term) is significantly different from 0, it means that the slopes of the sex-specific regression lines are divergent; the difference in LVM between girls and boys is significant and varies for different values of the body size parameter.

## Development of sex-specific and non-specific LVM normative data

In this part of the study, based on our study group, we developed seven sets of normative data of LVM. The sets were generated using different methodologies and body size parameters. For each set, normative data were generated first for a combined group of girls and boys, and next, separately for girls and boys. Thus, every set consisted of LVM normative data that were non-sex-specific and sex-specific. Then, using all these LVM normative data, for every study participant, seven pairs of LVM z-score values were calculated based on her/his actual LVM. Each pair consisted of LVM z-scores computed based on sex-specific normative data and LVM z-score calculated based on non-sex-specific normative data from the same set.

At first, three sets of LVM normative data, according to the LMS method [35], were developed with height, BSA, and cLBM, respectively, as explanatory variables. This procedure was applied previously by Foster et al. [29,36]. In the LMS method, based on the relationship between LVM and body size variable in the tested group, the expected mean LVM (M), coefficient of variation (S), and skewness (L) for each level of body size variable are generated. In our study, the corresponding L, M, and S values were developed first for the combined group, and then separately for girls and boys. For an individual child, the LVM z-scores were

calculated from the L, M, and S values corresponding to the child's body size parameter's magnitude, according to the equation:

$$z-score = \frac{\left[\left(\frac{actual\ LVM}{M}\right)^{L} - 1\right]}{L \times S}$$

Then, two sets of LVM normative data were computed based on the commonly used LVM indices—the ratio of LVM to BSA and the ratio of LVM to height raised to the power of 2.7, respectively. In this method, normative data are developed as a mean and standard deviation of the LVM indices for the tested group. In our study, after calculating individual LVM index for every participating child and adolescent, the mean and standard deviation was computed first for the combined group, and next separately for girls and boys. The LVM z-scores are calculated, according to the equation:

$$z-score = \frac{actual\ LVM\ index - mean\ LVM\ index\ (normative\ data)}{standard\ deviation\ (normative\ data)}$$

Next, one set was constructed based on the ratio of LVM to the allometrically adjusted BSA. The procedure proposed by Lopez et al. [26] was applied. Single allometric exponent specific for our combined study group—without division on sex was determined. The allometric equation was fitted for the bivariate relationship between LVM and BSA. This equation has the general form: $LVM = a(BSA)^{b}$, where $b$ is an allometric exponent. Logarithmic transformation gives the linearized form of this equation: $\ln(LVM) = \ln(a)+b\ln(BSA)$, allowing estimation of the allometric coefficients using linear least squares regression modeling [37]. The procedure of LVM normative data preparation and calculation of individual z-scores was the same as for the commonly used LVM indices.

Finally, one set was developed based on the ratio of LVM to the allometrically adjusted height according to the method proposed in our previous study [18]. Based on the bivariate relationship between LVM and height, an allometric exponent for the combined group and then two distinct allometric exponents for girls and boys were determined respectively. Then, corresponding LVM indices were calculated, and normative data for the combined and sex-specific groups were obtained, respectively. The LVM normative data preparation and calculation of individual z-scores were made in the same way as for the ratio of LVM to the allometrically adjusted BSA and the commonly used LVM indices. Examples of LVM z-score calculations from the L, M, and S values corresponding to the body size parameter's magnitude, and based on the mean and standard deviation of LVM indices, were presented in a supporting file in our previous article [18].

## Method for comparison of sex-specific and non-specific LVM normative data

All the comparisons were made between LVM z-scores computed based on sex-specific normative data and LVM z-scores calculated based on non-sex-specific normative data from the same set. We started with graphical presentations of the computed LVM z-scores on scatter plots. For each set of data, the sex-specific z-scores were displayed against the non-specific ones. Besides, a line of equality was drawn on the graph as well as one horizontal line and one vertical line at LVM z-score equal to +1.65, indicating the limit for diagnosis of LV hypertrophy [18,38].

Then, we examined whether the mean differences between the paired z-scores differ from 0. To check whether the mean differences differ between girls and boys, we used the

independent two-sample t-test. We have deepened this analysis and constructed scatter plots of the differences between non-specific and sex-specific z-score against the averages of non-specific and sex-specific z-scores. The scatter plots were similar to Bland-Altman plots [39]. However, the data for girls and boys were separated on each scatter plot. On these plots, we drew two horizontal lines corresponding to the mean difference for girls and boys, respectively. We fitted regression lines to the sex-specific data and tested for significance of slopes to verify whether the differences are uniform. The statistically significant slope indicates that the differences are not uniform.

To test the concordance between the LVM z-scores, we used contingency tables. Assuming that the LVM z-score above +1.65 indicates LV hypertrophy, we examined the percentage of discordant LV hypertrophy indications.

Analyses were made using IBM SPSS Statistics 25 (PS IMAGO PRO, Predictive Solutions, Poland) and LMS Chartmaker Pro (Medical Research Council, United Kingdom), For all statistical tests, a significance level of $\alpha = 0.05$ was used.

## Results

### The study participants' characteristics

The characteristics of the study participants are presented in Table 1. It shows the group of girls, boys, and the combined group, respectively, that were used for the evaluation of the effect of sex on the relationship between LVM and body size as well as the normative data development.

**Table 1. Characteristics of the study participants.**

|  | Girls | Boys | Combined groups |
|---|---|---|---|
| *n* | 331 | 490 | 821 |
| Age [years] | 12.0 (5.0) | 13.0 (5.0) | 12.0 (5.0) |
| Height [cm] | 153.0 (23.0) | 163 (34.0) | 158.0 (85.0) |
| Body mass [kg] | 41.8 (20.4) | 50.25 (30.8) | 46.3 27.2) |
| BSA [m$^2$] | 1.33 (0.41) | 1.51 (0.61) | 1.41 (0.56) |
| cLBM [kg] | 29.78 (13.25) | 38.41 (23.87) | 33.10 (20.38) |
| LVM [g] | 103.98 (44.01) | 126.42 (80.36) | 113.27 (62.57) |
| LVIDd [mm] | 42.0 (6.0) | 46.0 (9.0) | 44.0 (8.0) |
| IVSd [mm] | 8.0 (1.0) | 8.0 (2.0) | 8.0 (2.0) |
| PWTd [mm] | 7.0 (1.0) | 8.0 (2.0) | 8.0 (2.0) |
| RWT | 0.35 (0.05) | 0.36 (0.06) | 0.36 (0.06) |
| RHR [beats/min] | 75 (15) | 68 (15) | 71 (16) |
| SBP [mm/Hg] | 111 (19) | 116 (17) | 114 (18) |
| DBP [mm/Hg] | 64 (12) | 65 (10) | 64 (11) |
| Training [min] | 240 (180) | 270 (180) | 270 (180) |

Data are expressed as "median (interquartile range)"; BSA, body surface area according to the Haycock formula [33]; cLBM, lean body mass computed according to Foster's at al. equations [34]; LVM, left ventricular mass; LVIDd, left ventricular internal dimension; IVSd, interventricular septal thickness; PWTd, posterior wall thickness; RWT, relative wall thickness calculated as $RWT = 2 \times PWTd/LVIDd$ [2]; Training stands for the weekly volume of training. It is a measure of participation in sports activity and was estimated as the product of the average number of training sessions per week and the average duration of a single session; RHR, resting heart rate; SBP and DBP, systolic, and diastolic blood pressure, respectively.

## Effect of sex on the relationship between LVM and body size

The scatter plots of LVM against height, BSA, cLBM, and body mass, respectively, with sex-specific regression lines fitted to data points representing LVM of girls and boys, are shown in Fig 1. The slopes of the regression lines fitted to the data points corresponding to LVM of girls are different from the slopes of the regression lines for boys. For all the body size variables, the lines for boys have a higher slope than for girls, suggesting that sex affects the relationship between LVM and body size.

Table 2 presents the coefficients estimated upon the regression model that had been constructed to compare the $y$-intercepts and slopes of the regression lines. For all the analyzed relationships of LVM against body mass variable, the coefficient $\beta 3$ is significantly different from 0. Then, the slopes of the sex-specific regression lines are divergent, and this indicates that the difference in LVM between girls and boys is significant and varies with the body size parameter's magnitude.

The coefficient $\beta_2$ in the presented model is significantly different from 0 for all body size parameters except cLBM. Thus, only for cLBM, the $y$-intercepts of the sex-specific regression lines are not different. However, this effect is redundant in the presence of significant $\beta_3$.

## The sex-specific and non-specific LVM normative data

Three sets of LVM normative data, generated using the LMS method, are presented as the L, M, and S values corresponding to each level of height, BSA, and cLBM, respectively, in (S1, S2, and S3 Datasets, respectively). The means and standard deviations, presented in Table 3, are LVM normative data that were produced based on the LVM indices. The individual LVM z-scores for subsequent comparison were computed upon the L, M, and S values, as well as the means and standard deviations. Table 3 also shows the exponents that were used in the allometrically adjusted indices. These include exponents, which were estimated upon our data to adjust BSA and height.

## Comparison of sex-specific and non-specific LVM normative data

The scatter plots with the sex-specific LVM z-scores against the non-specific are presented in Figs 2 and 3. Fig 2 contains the z-scores calculated from the normative data generated using the LMS method. In Fig 3 the z-scores calculated on normative data based on LVM indices are shown.

In all the scatter plots, presenting LVM z-scores computed on seven different methods of LVM normalization, the data points of girls are separated from the points of boys. Both the points of girls and boys deviate from the equality line. The setting of the regression lines fitted to the respective data points helps to see this clearly. It suggests that the sex-specific LVM z-scores differ from the non-specific. The dependent t-test for paired samples confirms it. For all the pairs, the mean differences between the paired LVM z-scores significantly differ from 0. The results of the analysis are presented in Table 4.

The mean differences between the paired LVM z-scores in girls are negative, and boys are positive. The independent two-sample t-test has verified that they differ between girls and boys (S1 Table). But what is essential, it shows that non-specific normative data underestimate relative LVM in girls and overestimate it in boys. Besides, the setting of the regression lines in scatter plots in Figs 2 and 3 suggests that the differences increase with the increase of the z-score. It has been confirmed by significant slopes of another regression lines that were fitted separately for girls and boys to the differences between non-specific and sex-specific z-scores relative to the averages of non-specific and sex-specific z-scores (S2 Table). S1 and S2 Figs contain the scatter plots displaying these regression lines, as well as two horizontal lines corresponding to the mean difference for girls and boys, respectively.

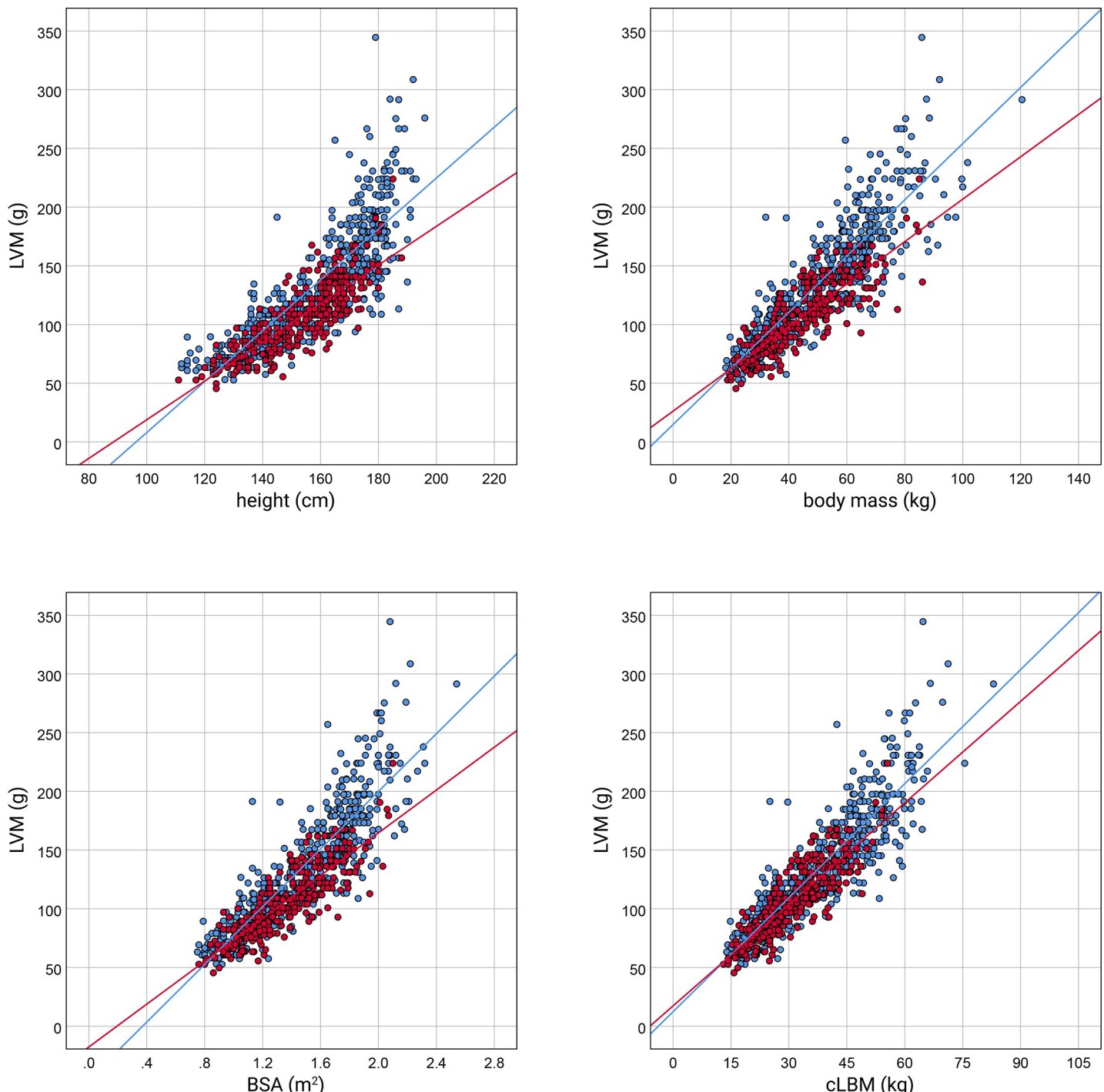

**Fig 1. The scatter plots of LVM against body size parameters in girls and boys.** On each scatter plot, the data points of girls (red) and boys (blue) are shown, and specific regression lines are fitted, respectively. BSA, body surface area according to the Haycock formula [33]; cLBM, lean body mass computed according to Foster's at al. equations [34];.

The horizontal and vertical lines on the scatter plots in Figs 2 and 3 at LVM z-score equal to +1.65, mark the limit for diagnosis of LV hypertrophy and allow seeing discordant LV hypertrophy indications. In all the scatter plots, the picture is the same—application of the LVM

**Table 2. Coefficients estimated in the applied regression model.**

| LVM *vs.* | $\beta_0$ | $\beta_1$ | $\beta_2$ | $\beta_3$ |
|---|---|---|---|---|
| **Height** | -145.8641 | 1.6473 | -62.9753 | 0.5205 |
| *p value* | *<0.0001* | *<0.0001* | *<0.0001* | *<0.0001* |
| **BSA** | -17.7802 | 91.1975 | -27.8703 | 31.5822 |
| *p value* | *= 0.0025* | *<0.0001* | *= 0.0001* | *<0.0001* |
| **cLBM** | 17.2105 | 2.8826 | -4.9287 | 0.3549 |
| *p value* | *<0.0001* | *<0.0001* | *0.3186* | *= 0.0152* |
| **Body mass** | 26.1783 | 1.8052 | -11.4169 | 0.5877 |
| *p value* | *<0.0001* | *<0.0001* | *= 0.0190* | *<0.0001* |

For the combined groups of girls and boys, the regression model has a form of the following equation: $y = \beta_0 + \beta_1 x + \beta_2 z + \beta_3 xz$, where $y$ is LVM, $x$ is body size variable, and $z$ is the dummy variable representing sex.

normative data that are not sex-specific underestimates relative LVM in girls and overestimates in boys.

We used contingency tables to analyze the concordance, and assuming that the LVM z-score above +1.65 indicates LV hypertrophy, we examined the percentage of discordant LV hypertrophy indications. The results of this analysis are presented in Table 5, and they confirm the picture from the scatter plots in Figs 2 and 3. The percentage of discordant indications, depending on the normalization method, ranges from 66.7% to 100% in girls and from 35.4% to 50% in boys.

## Discussion

The most important result of our study is a demonstration that normative data of left ventricular mass should be developed separately for girls and boys. Application of normative data that

**Table 3. The LVM normative data computed based on LVM indices.**

| | Allometric exponent | LVM index |
|---|---|---|
| **Girls** | | |
| LMV indexed to BSA | N/A | 77.4121 (11.1118) |
| LVM indexed to height$^{2.7}$ | 2.7 | 33.2248 (5.0743) |
| LVM indexed to BSA$^b$ | 1.3100 | 71.0274 (10.0675) |
| LVM indexed to height$^{bs}$ | 2.4340 | 37.1008 (5.5716) |
| **Boys** | | |
| LMV indexed to BSA | N/A | 90.0531 (17.3508) |
| LVM indexed to height$^{2.7}$ | 2.7 | 37.6220 (7.2802) |
| LVM indexed to BSA$^b$ | 1.3100 | 80.2761 (13.8976) |
| LVM indexed to height$^{bs}$ | 2.5776 | 39.7813 (7.6853) |
| **Combined groups** | | |
| LMV indexed to BSA | N/A | 84.9567 (16.3620) |
| LVM indexed to height$^{2.7}$ | 2.7 | 35.8492 (6.8284) |
| LVM indexed to BSA$^b$ | 1.3100 | 76.5473 (13.2882) |
| LVM indexed to height$^{bs}$ | 2.6217 | 37.1058 (7.0667) |

The LVM normative data are expressed as "mean (standard deviation)." For BSA$^b$, the BSA is raised to the power of $b$, where $b$ is equal to the allometric exponent estimated for the combined group; for height$^{bs}$, the height is raised to the power of $bs$, where $bs$ is equal to the allometric exponent that is group-specific—estimated separately for the combined group, for girls, and boys, respectively.

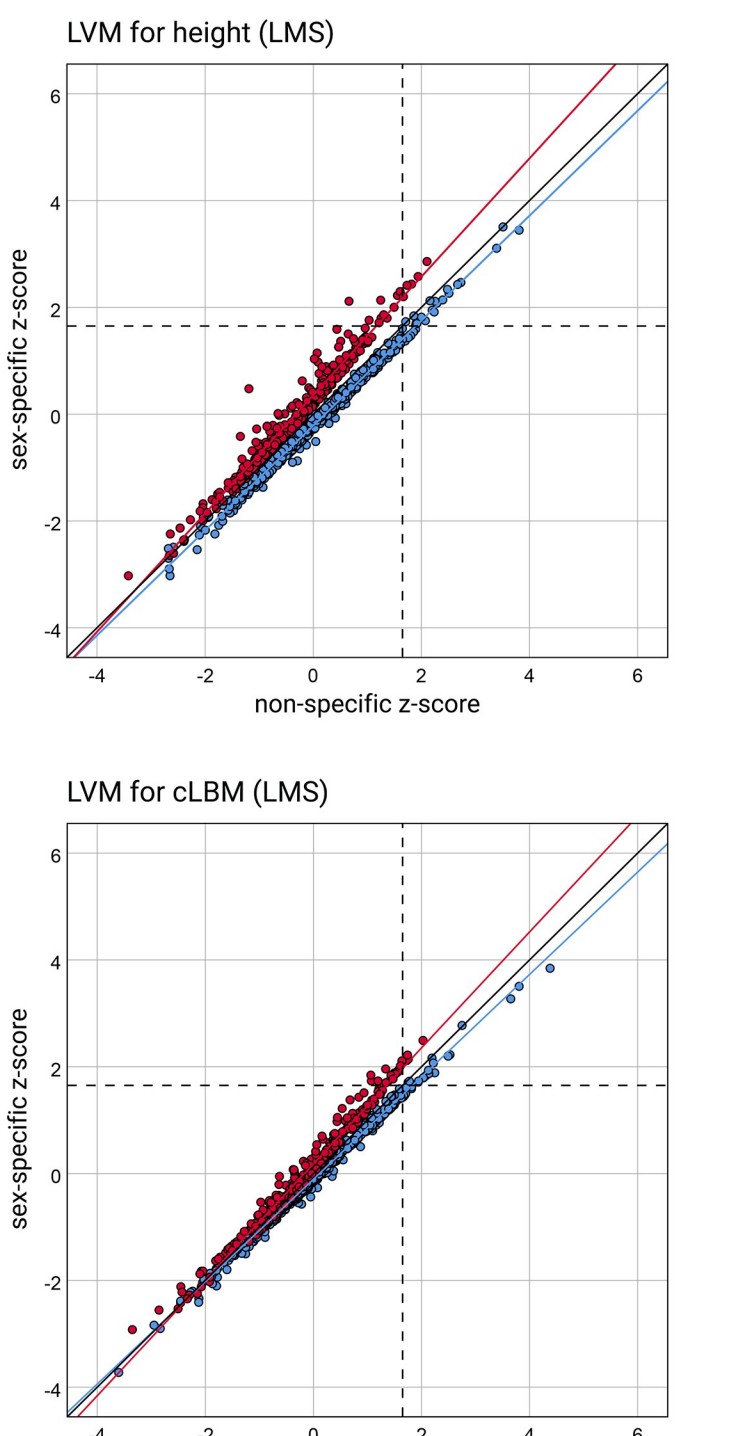

**Fig 2. Scatter plots of the LVM z-scores calculated from the normative data generated using the LMS method.** The sex-specific z-scores are displayed against the non-specific. The data points corresponding to girls are red, and to boys are blue. Regression lines are fitted to the sex-specific data—the solid red line to girls and the solid blue line to boys, respectively. The line of equality (solid black line) is drawn on each graph, as well as one horizontal line (dotted line) and one vertical line (dashed line) at LVM z-score equal to +1.65, indicating the limit for diagnosis of LV hypertrophy. BSA, body surface area according to the Haycock formula [33]; cLBM, lean body mass computed according to Foster's at al. equations [34];.

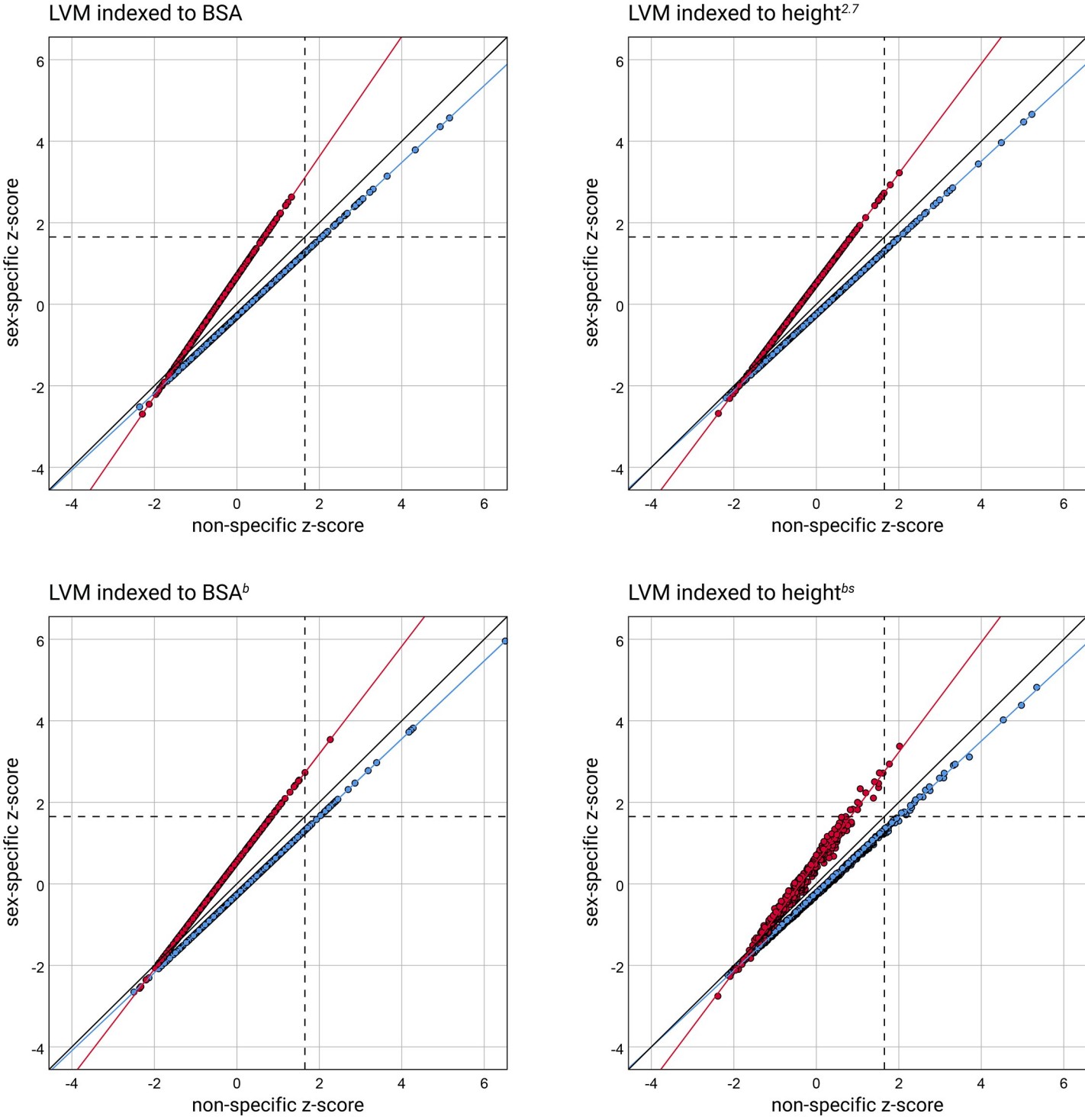

**Fig 3. The scatter plots of the z-scores calculated on normative data based on LVM indices–the sex-specific vs. non-specific.** The design of the scatter plots is the same as for Fig 2.

are generated on the combined group of girls and boys results in an underestimation of relative LVM in girls and overestimation in boys. From the clinical perspective, this increases the frequency of LV hypertrophy diagnosis in boys, but in girls, it may cause that LV hypertrophy is

**Table 4. The differences between non-specific and sex-specific LVM z-scores.**

|  | Mean difference | p-value |
|---|---|---|
| **Girls** |  |  |
| LVM for Height (LMS) | -0.3218 (0.2184) | p<0.0001 |
| LVM for BSA (LMS) | -0.3725 (0.2653) | p<0.0001 |
| LVM for cLBM (LMS) | -0.1670 (0.1608) | p<0.0001 |
| LMV indexed to BSA | -0.4611 (0.3209) | p<0.0001 |
| LVM indexed to height$^{2.7}$ | -0.3843 (0.2569) | p<0.0001 |
| LVM indexed to BSA[b] | -0.4154 (0.2438) | p<0.0001 |
| LVM indexed to height[bs] | -0.3943 (0.2858) | p<0.0001 |
| **Boys** |  |  |
| LVM for Height (LMS) | 0.2171 (0.0956) | p<0.0001 |
| LVM for BSA (LMS) | 0.2507 (0.0622) | p<0.0001 |
| LVM for cLBM (LMS) | 0.1118 (0.0845) | p<0.0001 |
| LMV indexed to BSA | 0.3115 (0.0604) | p<0.0001 |
| LVM indexed to height$^{2.7}$ | 0.2596 (0.0662) | p<0.0001 |
| LVM indexed to BSA[b] | 0.2806 (0.0459) | p<0.0001 |
| LVM indexed to height[bs] | 0.2664 (0.0730) | p<0.0001 |

The data are expressed as "mean difference (standard deviation)." LMS in brackets means that these LVM normative data were produced using the LMS method. For BSA[b], the BSA is raised to the power of *b*, where *b* is equal to the allometric exponent estimated for the combined group; for height[bs], the height is raised to the power of *bs*, where *bs* is equal to the allometric exponent that is group-specific—estimated separately for the combined group, for girls, and boys, respectively.

unrecognized. This finding is consistent with the result of the evaluation of the relationship between LVM and body size we have made. The course of changes of LVM relative to body size during development is different in girls and boys. The regression line for the relationship

**Table 5. The number of indications of LV hypertrophy based on sex-specific and non-specific LVM normative data.**

|  | Sex-specific normative data | Non-specific normative data | Percent of discordant indications (95% CI) |
|---|---|---|---|
| **Girls** |  |  |  |
| LVM for Height (LMS) | 15 | 5 | 66.7% (38.3–88.2%) |
| LVM for BSA (LMS) | 19 | 1 | 94.7% (73.9–99.9%) |
| LVM for cLBM (LMS) | 21 | 5 | 76.2% (52.8–91.8%) |
| LMV indexed to BSA | 20 | 0 | 100% (83.2–100%) |
| LVM indexed to height$^{2.7}$ | 16 | 2 | 87.5% (61.6–98.5%) |
| LVM indexed to BSA[b] | 19 | 2 | 89.5% (66.9–98.7%) |
| LVM indexed to height[bs] | 15 | 2 | 86.7% (59.5–98.3%) |
| **Boys** |  |  |  |
| LVM for Height (LMS) | 20 | 31 | 35.4% (19.2–54.6%) |
| LVM for BSA (LMS) | 19 | 33 | 42.4% (25.5–60.8%) |
| LVM for cLBM (LMS) | 18 | 28 | 35.7% (18.6–55.9%) |
| LMV indexed to BSA | 29 | 50 | 42.0% (28.2–56.8%) |
| LVM indexed to height$^{2.7}$ | 22 | 42 | 47.6% (32.0–63.6%) |
| LVM indexed to BSA[b] | 26 | 50 | 48.0% (33.7–62.6%) |
| LVM indexed to height[bs] | 22 | 44 | 50.0% (34.6–65.4%) |

The subjects were classified as having LVH when their LVM z-score > +1.65. Confidence intervals (CI) for the proportions are Clopper-Pearson exact confidence intervals. The designations of LVM normalization methods are the same as in Table 4.

between LVM and body size is steeper in boys comparing to girls. It means that for a given body size LVM in boys is higher than in girls.

Such an analysis of the relationship between LVM and body size has been made previously by others [27,28,30,40–43], and it seems, that there is an agreement that when considering the relationship between LVM and the elementary body size parameters, like body mass and height, the courses of changes of LVM in relation to these parameters are different in girls and boys. The difference becomes evident at puberty, and after puberty, boys definitively have higher LVM comparing to girls [42]. Adult men have higher unindexed LVM than women [44]. In athletes, both adolescents and adults, the pattern is the same—male athletes have higher LVM comparing to female athletes [14,15,45].

According to some researchers, there is no difference between boys and girls in the course of changes of LVM against LBM. They argue that since lean body mass (or fat-free mass, FFM) is the strongest determinant of LVM, the sex-related differences in LVM can be explained by the differences in LBM between boys and girls [41,46]. Similar suggestions were made for adult males and females, including athletes [44,47].

Recognizing LBM as a pivotal physiological determinant of LVM made this body size parameter potentially optimal for cardiac size scaling [29,32,46]. Since LBM cannot be measured directly, advanced indirect methods such as double X-ray absorptiometry, computed tomography, magnetic resonance imaging, or bioelectrical impedance analysis are required for reliable measurements. These measurements are not routinely available in the cardiac imaging laboratory, so researchers seek a surrogate parameter for LBM by allometrically transforming height, for example [32,42,48]. The most often used is the height raised to the power of 2.7 [48]. For LVM normalization, the LBM or FFA predictive equations, based on the elementary body size variables and their derivatives, are also proposed [34,46].

In our study, we evaluated not only the relationship between LVM and body mass, height, and BSA but also between LVM and LBM computed based on the predictive equations introduced by Foster et al. [34]. For all the four body size parameters, including the computed LBM, there was a significant difference between the slopes of the lines fitted to the LVM data points of girls and boys, respectively.

Daniels et al. [32] and de Simone et al. [42] claimed that after proper normalization, the relationship between the normalized LVM and body size is not statistically significant, and normative data can be produced without division on sex. Yet the results of our study show that if we want to avoid errors when diagnosing LV hypertrophy, we should use sex-specific normative data. It is consistent with the indication of Pela et al. [43], who recommend sex-specific normative data in the cardiovascular screening of adolescent athletes.

There are studies where the sex-specific LVM normative data for the pediatric population were developed because the authors noted a significant difference in relative LVM between boys and girls [27–30]. However, in many others, the LVM normative data were constructed without division on sex. Pettersen et al. [21] produced echocardiographic normative data for the combined group of girls and boys and made no statements on why they did not take into account the potential sex differences. Similarly did Kampmann et al. [22]. In other studies, the authors evaluated the influence of sex on the result of the measurement and found no significant effect [23–25]. Lopez et al. [26] found a significant statistical impact but recognized it as not significant from a clinical perspective. They argued that the differences between $R^2$ in the regression models with sex and $R^2$ in the models without sex were small, and comparisons of echocardiographic dimensions that were predicted based on these models, exercised on two hypothetical boys, had shown small differences between the models.

Therefore, in the crucial part of our study, we developed normative data using different methods and different body size parameters as explanatory variables. We developed sets

containing pairs of sex-specific and non-specific LVM normative data and compared them mutualy within the sets. The results of that comparison confirmed this intuitive indication that emerged after the analysis of the relationship between LVM and the body size parameters: If we want to avoid errors when diagnosing LV hypertrophy, we should use sex-specific normative data, regardless of the body size parameter used as the explanatory variable. Examples of LVM z-scores calculations are presented in the (S1 Text). They picture the overestimation of relative LVM in boys and underestimation in girls when the LVM normative data that are not sex-specific are used.

The clinical perspective seems to be particularly important here because, as we have shown, choosing LVM normative data that are not sex-specific can result in LVH not being identified in an adolescent girl who practices sport. Consequently, the etiology of hypertrophy will not be differentiated and proper medical management, required in case of diagnosed cardiac pathology, not introduced. This approach increases cardiac risk in girls participating in sport. In turn, relying on non-specific normative data in medical evaluation of boy practicing sports can lead to a false-positive diagnosis of LVH. Unnecessary measures are then introduced, such as exclusion from sport and unjustified additional clinical tests that can cause anxiety in the boy and his family.

In everyday clinical practice, only LV wall thickness measurements are often used to identify and monitor LV hypertrophy. Still, it should be noted that the sensitivity, specificity, and prognostic accuracy of LV mass in detecting LV hypertrophy are higher than when measuring only LV wall thickness [4,49]. However, left ventricular mass should be appropriately normalized for body size. It is particularly important in children and adolescents, because of high variability in height and body mass, even among similar aged children.

The study participants were child and adolescent athletes. The athletic population may be considered special because regular exercise contributes to an increase in cardiac size [15], and specific LVM normative data are recommended for child and adolescent athletes [20]. A question may thus arise as to whether the results of the study should be applied to all children and adolescents.

The increase in cardiac size in response to exercise is an adaptive phenomenon linked to the improvement of exercise capacity. Not only athletes but generally all healthy children, both boys, and girls, have higher LVM when their exercise capacity is higher [50]. However, boys have higher exercise capacity than girls of the same age, and this is true for both athletic and non-athletic populations [51]. Perhaps in the youngest, the difference is not seen, but after the age of about twelve, it becomes significant [52]. The same pattern is observed in the case of LVM. Before puberty, LVM in healthy boys and girls is similar, and at puberty, the difference between boys and girls becomes evident, with higher LVM in boys [53].

Thus, higher exercise capacity is associated with higher LVM, and within comparable groups of healthy children, especially adolescents, boys have higher exercise capacity and LVM than girls. That is not solely specific to athletes. Normative data for LVM should be developed separately for boys and girls for all children and adolescents, regardless of whether they practice sports or not.

## Study limitations

Our study has limitations. Since the study was retrospective, based on historical medical records that had been collected since 2013, the intraobserver and interobserver variability for echocardiographic measurements were not analyzed. However, all these echocardiographic measurements were performed by two experienced cardiologists in one medical center.

The same groups of girls and boys were used for LVM normative data development and further comparison, and it can be considered a limitation. It might seem that comparing LVM z-scores in a group that was previously used to produce LVM normative data, upon which the z-scores were then calculated, introduces bias by reducing variability. However, if it was true, it would decrease the differences between the paired z-scores and improved concordance. The procedure is statistically valid, and the comparison made in the same group rather strengthens the significance of the results.

We used an ethnically homogenous group of child and adolescent athletes from 5 to 19 years of age for this analysis. It might be argued that such group characteristics limit the possibility of generalization. We do not question the necessity of further research to confirm the results in younger children, adults, and subjects from different ethnic groups. Yet still, we are sure that the results are reliable, good in quality, and useful.

Our analysis did not take into account other factors than body size and sex, which potentially influence LVM, like blood pressure, heart rate, or fat mass. However, the participants of the study were all healthy child and adolescent athletes, under regular medical monitoring. Thus, all pathological factors were excluded, and blood pressure and heart rate were in the physiological range. In such a situation, their influence on LVM is minimal, although statistically significant [30]. It does not interfere significantly with the effect of body size and sex on LVM.

The children and adolescents whose echocardiographic data were used in this study were athletes. As the training volume and intensity have to be adapted to the athlete exercise capacity, we cannot exclude that the different intensity of training in girls and boys additionally contributed to the fact that for a given body size, LVM in boys was higher than in girls. It may raise a concern about the application of the results to all pediatric populations. Although we agree that the specificity of the population examined in this study may have influenced the results, we are convinced that athletic training might only amplify the already existing differences in LVM between boys and girls. Therefore, for all pediatric populations, one should use LVM normative data that were developed separately for girls and boys.

It should be noted that although it was not the purpose of the study to present normative data of LVM for youth athletes, we have performed controlling procedure for all the developed LVM normative data. The effectiveness of the normalization procedures was tested in terms of whether body size information was eliminated in the generated normative data. Relationships between the calculated LVM z-scores and the corresponding body size variables were analyzed. The Pearson correlation coefficient and the slope of the linear regression line were examined [37].

The LVM normative data produced based on a simple index of LVM to BSA do not meet the statistical criteria for effective normalization. The presence of the relationship between LVM z-scores and BSA has been confirmed for both sex-specific and non-specific normative data. The Pearson correlation coefficients and the slopes of the linear regression lines are statistically significant. These coefficients are also significant for LVM indexed to height raised to the power of 2.7 and LVM indexed to allometrically adjusted BSA, for sex-specific normative data of girls.

It is consistent with the results of our previous studies [17,18], and the presence of a significant relationship between LVM indexed to height raised to the power of 2.7 and height in girls additionally confirms the results of the current work.

The Pearson correlation coefficients and the slopes of the linear regression lines, which were examined to test whether body size information was eliminated in the generated normative data, are presented in S3 Table.

## Conclusions

The study was designed to explore the effect of sex on the relationship between LVM and body size variables used in the normalization of cardiac size and to test concordance between sex-specific and non-specific LVM normative data developed according to different methods. The primary purpose of the study was to answer the question of whether LVM normative data should be sex-specific. The study showed that in child and adolescent athletes from 5 to 19 years of age, the course of changes of LVM relative to body size during development is different in girls and boys and that for a given body size LVM in boys is higher than in girls. Application of normative data that are not sex-specific results in an underestimation of relative LVM in girls and overestimation in boys. From the clinical perspective, this increases the frequency of LV hypertrophy diagnosis in boys, but in girls, it may cause that LV hypertrophy is unrecognized. Therefore, if we want to avoid errors when diagnosing left ventricular hypertrophy in children and adolescents, we should use normative data for left ventricular mass that were developed separately for girls and boys, regardless of the body size parameter used as the explanatory variable.

## Supporting information

**S1 Dataset. The sets of the L, M, and S values corresponding to each level of height.**
(TXT)

**S2 Dataset. The sets of the L, M, and S values corresponding to each level of BSA.**
(TXT)

**S3 Dataset. The sets of the L, M, and S values corresponding to each level of cLBM.**
(TXT)

**S4 Dataset. The original dataset.**
(TXT)

**S1 Table. The mean differences between the paired non-specific and sex-specific z-scores in girls and boys.**
(DOCX)

**S2 Table. Pearson correlation coefficients and the slopes of the regression lines for relationships between the differences between non-specific and sex-specific z-score and the averages of non-specific and sex-specific z-scores.**
(DOCX)

**S3 Table. Pearson correlation coefficients and the slopes of the regression lines for relationships between the LVM z-scores and the corresponding body size variables.**
(DOCX)

**S1 Fig. Scatter plots of the differences between non-specific and sex-specific z-scores relative to the averages of non-specific and sex-specific z-scores for the z-scores calculated from the normative data generated using the LMS method.** The data points corresponding to girls are red, and to boys are blue. Regression lines are fitted to the data points—the solid red line to girls and the solid blue line to boys. Two horizontal lines corresponding to the mean difference for girls (dashed red line) and boys (dashed blue line) are drawn as well. BSA, body surface area according to Haycock formula [33]; cLBM, lean body mass computed according to Foster's at al. equations [34];
(TIF)

**S2 Fig. Scatter plots of the differences between non-specific and sex-specific z-scores relative to the averages of non-specific and sex-specific z-scores calculated upon the normative data based on LVM indices.** The design of the scatter plots is the same as for S1 Fig.
(TIF)

**S1 Text. Examples of LVM z-score calculations.** They picture the overestimation of relative LVM in boys and underestimation in girls when the LVM normative data that are not sex-specific are used.
(DOCX)

## Author Contributions

**Conceptualization:** Hubert Krysztofiak.

**Data curation:** Hubert Krysztofiak, Marcel Młyńczak, Andrzej Folga, Wojciech Braksator.

**Formal analysis:** Hubert Krysztofiak, Marcel Młyńczak.

**Funding acquisition:** Hubert Krysztofiak.

**Investigation:** Hubert Krysztofiak, Łukasz A. Małek, Andrzej Folga, Wojciech Braksator.

**Methodology:** Hubert Krysztofiak, Marcel Młyńczak, Łukasz A. Małek.

**Project administration:** Hubert Krysztofiak.

**Resources:** Hubert Krysztofiak.

**Software:** Hubert Krysztofiak.

**Supervision:** Hubert Krysztofiak.

**Validation:** Hubert Krysztofiak.

**Visualization:** Hubert Krysztofiak.

**Writing – original draft:** Hubert Krysztofiak.

**Writing – review & editing:** Hubert Krysztofiak, Marcel Młyńczak, Łukasz A. Małek, Andrzej Folga, Wojciech Braksator.

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
