## [Decision Letter · Decision Letter 0]

16 Mar 2020

PONE-D-20-05310

Left ventricular mass normalization in child and adolescent athletes must account for sex differences.

PLOS ONE

Dear Dr Krysztofiak,

Thank you for submitting your manuscript to PLOS ONE. After careful consideration, we feel that it has merit but does not fully meet PLOS ONE’s publication criteria as it currently stands. Therefore, we invite you to submit a revised version of the manuscript that addresses the points raised during the review process.

Please address all the issues raised by the reviewers before re-submission.

We would appreciate receiving your revised manuscript by Apr 30 2020 11:59PM. To enhance the reproducibility of your results, we recommend that if applicable you deposit your laboratory protocols in protocols.io, where a protocol can be assigned its own identifier (DOI) such that it can be cited independently in the future. For instructions see: http://journals.plos.org/plosone/s/submission-guidelines#loc-laboratory-protocols

We look forward to receiving your revised manuscript.

Kind regards,

Elena Cavarretta, M.D., Ph.D.

Academic Editor

PLOS ONE

Journal Requirements:

Reviewers' comments:

Reviewer's Responses to Questions

**Comments to the Author**

1. Is the manuscript technically sound, and do the data support the conclusions?

Reviewer #1: Yes

Reviewer #2: Yes

2. Has the statistical analysis been performed appropriately and rigorously? 

Reviewer #1: Yes

Reviewer #2: Yes

3. Have the authors made all data underlying the findings in their manuscript fully available?

Reviewer #1: Yes

Reviewer #2: Yes

4. Is the manuscript presented in an intelligible fashion and written in standard English?

Reviewer #1: Yes

Reviewer #2: Yes

5. Review Comments to the Author

Reviewer #1: I had the opportunity to review the study entitled “Left ventricular mass normalization in child and adolescent athletes must account for sex differences” by Hubert Krysztofiak and colleagues. In this study they analyzed 331 girls and 490 boys between 5 and 19 years old to assess the effect of sex on Left Ventricular Mass (LVM), based on echocardiography. Authors developed seven sets of the LVM normative data using different methodologies to test concordance between sex-specific and non-specific normative data, and they found that data should be developed separately for girls and boys practicing sports to avoid underdiagnosing left ventricular hypertrophy in girls and to avoid overdiagnosis in boys. The manuscript is well written and is presented in an intelligible standard English. Despite the difficulty to understand how the normative data sets have been created, the reader can understand it more easily, thanks to the explanations given step by step. Although the question of whether LVM is sex- specific or not is still open, results from this manuscript are in according to other manuscripts that have reached the same conclusion. I believe that despite the difficulty in understanding the normative data set, the methodology used to test the hypothesis has been adequate to achieve the results.

However, I would like two topics to be developed a little more extensively.

First, in the discussion section, what does the measurement of LVM provide on the determination of the thickness of the interventricular septum and posterior wall in the determination of left ventricular hypertrophy?

Second on page 21, line 21, I would like the clinical application to be exemplified in some way. For example, the determination of Z score from an individual athlete using the normative data of the supplementary material.

Reviewer #2: The article by Krysztofiak H. aims to assess whether the left ventricular mass (LVM) normative echocardiographic data should be developed separately for girls and boys practicing sports. By using a linear regression model, the authors demonstrated that the sex affects the relationship between LVM and body size in a large cohort of Caucasian pediatric subjects aged 5-19 years old, suggesting the claim for sex-specific LVM normative data.

In addition to the limitations already mentioned in the article, there are some concerns that, in my opinion, should be addressed by the authors in order to improve the clinical relevance of the paper:

- How long did it take to enroll the subjects?

- The type of sport might differentially affect LV remodeling. Do you have any data on the discipline practiced by the enrolled population? Were all endurance sports?

- The authors should also report the intensity (low power or high power) and duration (hours per week) of sport activity and whether the subjects were professional or amateur-level athletes.

- Can you exclude different levels of training in the two groups?

- Did you exclude young athletes with abnormalities on ECG suggesting cardiomyopathies but minor/ambiguous abnormalities on TTE?

- Do you have any electrocardiographic data about the population investigated? It would be interesting to know how many athletes had positive criteria for LVH at ECG among the two groups in contrast.

- Would it be methodologically finer to gather patients in specific age-groups?

- Can you provide age and sex-related LVM reference values?

- Can you provide intra-observer and inter-observer analyses to guarantee for echocardiographic measurement reproducibility?

- Mean blood pressure and heart rate values should be reported.

6. PLOS authors have the option to publish the peer review history of their article (what does this mean?). If published, this will include your full peer review and any attached files.

Reviewer #1: Yes: Marc Abulí Lluch

Reviewer #2: No

---

## [Author Response · Author response to Decision Letter 0]

27 Mar 2020

March 31st, 2020

Elena Cavarretta, MD, PhD.

Academic Editor

PLOS ONE

Dear Professor Cavarretta,

According to your decision, after taking into account the concerns and criticisms addressed by the reviewers, we are submitting a revised version of the manuscript entitled " Left ventricular mass normalization in child and adolescent athletes must account for sex differences " (PONE-D-20-05310). Considering the reviewers' suggestions, we have introduced a few changes to the manuscript.

Below are detailed, point-by-point answers (written in blue) to the comments and queries from the reviewers (red). Quotes from the manuscript are italicized.

On behalf of all co-authors, sincerely yours,

Hubert Krysztofiak, MD, PhD

orcid.org/0000-0003-4567-1433

Mossakowski Medical Research Centre

Polish Academy of Sciences

5 Pawinskiego Str., 02-106 Warsaw, Poland

hkrysztofiak@imdik.pan.pl

 

Response to reviewers

Reviewer #1: I had the opportunity to review the study entitled “Left ventricular mass normalization in child and adolescent athletes must account for sex differences” by Hubert Krysztofiak and colleagues. In this study they analyzed 331 girls and 490 boys between 5 and 19 years old to assess the effect of sex on Left Ventricular Mass (LVM), based on echocardiography. Authors developed seven sets of the LVM normative data using different methodologies to test concordance between sex-specific and non-specific normative data, and they found that data should be developed separately for girls and boys practicing sports to avoid underdiagnosing left ventricular hypertrophy in girls and to avoid overdiagnosis in boys. The manuscript is well written and is presented in an intelligible standard English. Despite the difficulty to understand how the normative data sets have been created, the reader can understand it more easily, thanks to the explanations given step by step. Although the question of whether LVM is sex- specific or not is still open, results from this manuscript are in according to other manuscripts that have reached the same conclusion. I believe that despite the difficulty in understanding the normative data set, the methodology used to test the hypothesis has been adequate to achieve the results.

However, I would like two topics to be developed a little more extensively.

Q1: First, in the discussion section, what does the measurement of LVM provide on the determination of the thickness of the interventricular septum and posterior wall in the determination of left ventricular hypertrophy?

Thank you very much for the comment.

Calculated left ventricular mass is a derivative of the septal and posterior wall thickness as well as LV internal diameter. Although the use of LV wall thickness measurements to identify and monitor LV hypertrophy may seem easier in everyday clinical practice, it should be noted that the sensitivity, specificity, and prognostic accuracy of LV mass in detecting LV hypertrophy are higher than when measuring only LV wall thickness. Please, see Barbieri A, et al. Left ventricular hypertrophy reclassification and death: application of the Recommendation of the American Society of Echocardiography/European Association of Echocardiography. 

We have briefly addressed this issue at the end of the Discussion section.

Q2: Second on page 21, line 21, I would like the clinical application to be exemplified in some way. For example, the determination of Z score from an individual athlete using the normative data of the supplementary material.

We have added examples of LVM z-scores calculations to the supporting information. In the supplementary file (S1 Text), the LVM z-scores calculations are shown. They picture the overestimation of relative LVM in boys and underestimation in girls when the LVM normative data that are not sex-specific are used.

The relevant information was included in the Discussion.

Reviewer #2: The article by Krysztofiak H. aims to assess whether the left ventricular mass (LVM) normative echocardiographic data should be developed separately for girls and boys practicing sports. By using a linear regression model, the authors demonstrated that the sex affects the relationship between LVM and body size in a large cohort of Caucasian pediatric subjects aged 5-19 years old, suggesting the claim for sex-specific LVM normative data.

In addition to the limitations already mentioned in the article, there are some concerns that, in my opinion, should be addressed by the authors in order to improve the clinical relevance of the paper:

Q3: How long did it take to enroll the subjects?

It was a retrospective study based on echocardiographic data collected between 2013 and 2018. We have added this information to the sub-section "The study participants".

Q4: The type of sport might differentially affect LV remodeling. Do you have any data on the discipline practiced by the enrolled population? Were all endurance sports?

Yes, we have information about the sport practiced by our youth athletes. Starting with the most popular, they had practiced: soccer, swimming, basketball, handball, fencing, rowing, tennis, dancing, distance running, speed skating, cycling, sailing, and martial sports like karate, taekwondo, judo, and wrestling.

As you can see, we studied a group of young athletes practicing various sports. However, since these were child and adolescent athletes, predominantly amateur, the training was focused mainly on general physical development, building the aerobic capacity, and motor skills.

We have added this information to the sub-section "The study participants".

Q5: The authors should also report the intensity (low power or high power) and duration (hours per week) of sport activity and whether the subjects were professional or amateur-level athletes.

The children and adolescents whose echocardiographic data were used in this study were athletes practicing amateur-level sport, mainly. Their training was focused primarily on general physical performance, building the aerobic capacity, and motor skills.

At the medical evaluation, we had been registering the weekly volume of training estimated as the product of the average number of training sessions per week and the average duration of a single session. We have added the information about training volume to Table 1, which presents the characteristics of the study participants.

It seems important to note that the main aim of our study was not a presentation of LVM normative data for a specific group of children and adolescents. This is a continuation of our previous works on the improvement of the methodology of LVM scaling, and these normative data, which were constructed in the present study, were used to evaluate the concordance between sex-specific and non-specific LVM normative data developed according to different methods.

Q6: Can you exclude different levels of training in the two groups?

A good question. We cannot exclude that in our study group, the level of training is different in girls and boys. Even if the volume of training would be similar, it is almost for sure that the absolute exercise intensities differ. Perhaps in the youngest, the differences are nonsignificant, but with age, the gap is growing. A comparison of world records shows it clearly. There are differences in exercise capacity between boys and girls, which cannot be explained by differences in body size. Of course, the training volume and intensity have to be adapted to the athlete exercise capacity, and in general, the absolute training workloads in girls are lower. However, in the context of the question posed in this study, these physiological differences further support the conclusions of the study that normative data for left ventricular mass should be developed separately for girls and boys.

We have addressed this issue in the sub-section "The study limitations."

Q7: Did you exclude young athletes with abnormalities on ECG suggesting cardiomyopathies but minor/ambiguous abnormalities on TTE?

Yes, certainly. Our study group consisted of healthy children and adolescents undergoing regular medical monitoring as part of the periodic preparticipation physical evaluation (PPE) at the National Center for Sports Medicine. In our country, periodic PPE is mandatory for all athletes. For children and adolescents, it is delivered as a part of the public health service. When during PPE the examining physician noticed innocent heart murmurs or suspicious electrocardiographic findings, he or she ordered echocardiography. However, as we stated in "The study participants" sub-section, "the athletes in whom echocardiography revealed significant acquired or congenital heart diseases, affecting heart size and hemodynamics, were not included in the study."

Q8: Do you have any electrocardiographic data about the population investigated? It would be interesting to know how many athletes had positive criteria for LVH at ECG among the two groups in contrast.

We agree it would be interesting. The electrocardiogram is an obligatory part of the medical screening of athletes, and we are collecting ECG data. However, since the aim of the present study was to assess whether the LVM normative data should be developed separately for girls and boys practicing sports, a distinct investigation is needed for such analysis.

Q9: Would it be methodologically finer to gather patients in specific age-groups?

Considering the methodology of normalization of cardiac dimensions and LVM for body size in children and adolescents, we are sure that it would not improve the quality of the study. In our study, we applied methodologies introduced by others (see, for example, Foster et al. 2008, 2013, 2016; Pettersen et al. 2008, Lopez et al. 2017) and by our team in previous works. One needs to realize that we (like many others) normalize LVM for body size. Since in healthy children, body size changes with age in a predictable way, the body size parameters, and in our opinion, especially height, can be treated as a surrogate of age. When proper normalization for body size is made in children and adolescents, there is no need for additional division on age groups. Our previous works' primary aim, as well as the present study, is to improve the methodology of cardiac size scaling for body size.

Q10: Can you provide age and sex-related LVM reference values?

Regarding this and previous question, we would like to emphasize that the main aim of our study was not a presentation of LVM normative data for the group of child and adolescent athletes. The study aimed to assess whether, for reliable evaluation of LVM in children and adolescents, left ventricular mass normative data should be developed separately for girls and boys. These normative data, which were constructed in the present study, were used to evaluate the concordance between sex-specific and non-specific LVM normative data developed according to different methods.

In our previous studies, we presented normative data for child and adolescent athletes and compared them to that shown by others:

Krysztofiak H, Małek ŁA, Młyńczak M, Folga A, Braksator W (2018) Comparison of echocardiographic linear dimensions for male and female child and adolescent athletes with published pediatric normative data. PLoS ONE 13(10): e0205459. https://doi.org/10.1371/journal.pone.0205459

Krysztofiak H, Młyńczak M, Folga A, Braksator W, Małek ŁA. Normal Values for Left Ventricular Mass in Relation to Lean Body Mass in Child and Adolescent Athletes. Pediatr Cardiol (2019) 40: 204. https://doi.org/10.1007/s00246-018-1982-9

Yet the present normative data were made with care and using a reliable methodology. The LVM normative data developed according to the LMS method are usually presented as centile curves of LVM against explanatory body size variable, similar to the so popular growth curves. It is not practiced to present the curves for a narrow age range if they were constructed for a wide age range (see Foster et al. 2013 and 2016).

For the LVM normative data developed based on the ratio of LVM to the height adjusted with (raised to the power of) specific allometric exponents, these specific exponents were determined for full age range groups (5-19); "specific" means that it is distinctive for a particular group. Thus, it would not be reasonable to present normative data for narrower age sub-groups that were selected from the main group; if so, then these sub-groups would require their own specific allometric exponents. The same for the LVM normative data developed based on the ratio of LVM to the allometrically adjusted BSA.

Eventually, only for the LVM normative data constructed based on the ratios of LVM to BSA and LVM to height raised to the power of 2.7, the age ranges normative data would be justified. However, in our previous study, we proved that BSA is not a proper explanatory variable for cardiac size scaling (Krysztofiak et al. 2019-1). In another previous study, we showed that group-specific allometric exponent should be used instead of the universal exponent of 2.7, to avoid constraints related to incomplete elimination of body size information from the normalized LVM (Krysztofiak et at. 2019-2).

Q11: Can you provide intra-observer and inter-observer analyses to guarantee for echocardiographic measurement reproducibility?

Since the study was retrospective, based on historical medical records that had been collected since 2013, the intraobserver and interobserver variability for echocardiographic measurements were not analyzed. However, all these echocardiographic measurements were performed by two experienced cardiologists in one medical center.

We have added this statement to "The study limitations" sub-section.

Q12: Mean blood pressure and heart rate values should be reported.

Thank you for that recommendation, we have added the data to Table 1, which shows the characteristics of the study participants.

---

## [Decision Letter · Decision Letter 1]

4 May 2020

PONE-D-20-05310R1

Left ventricular mass normalization in child and adolescent athletes must account for sex differences.

PLOS ONE

Dear Dr Krysztofiak,

Thank you for submitting your manuscript to PLOS ONE. After careful consideration, we feel that it has merit but does not fully meet PLOS ONE’s publication criteria as it currently stands. Therefore, we invite you to submit a revised version of the manuscript that addresses the points raised during the review process.

Both Reviewers and myself believe the manuscript to be much improved from the previous iteration, however a number of issues remain, as outlined below, from Reviewer 2.

We would appreciate receiving your revised manuscript by Jun 18 2020 11:59PM. To enhance the reproducibility of your results, we recommend that if applicable you deposit your laboratory protocols in protocols.io, where a protocol can be assigned its own identifier (DOI) such that it can be cited independently in the future. For instructions see: http://journals.plos.org/plosone/s/submission-guidelines#loc-laboratory-protocols

We look forward to receiving your revised manuscript.

Kind regards,

Daniel M. Johnson, PhD

Academic Editor

PLOS ONE

Reviewers' comments:

Reviewer's Responses to Questions

**Comments to the Author**

1. If the authors have adequately addressed your comments raised in a previous round of review and you feel that this manuscript is now acceptable for publication, you may indicate that here to bypass the “Comments to the Author” section, enter your conflict of interest statement in the “Confidential to Editor” section, and submit your "Accept" recommendation.

Reviewer #1: All comments have been addressed

Reviewer #2: All comments have been addressed

2. Is the manuscript technically sound, and do the data support the conclusions?

Reviewer #1: Yes

Reviewer #2: Yes

3. Has the statistical analysis been performed appropriately and rigorously? 

Reviewer #1: Yes

Reviewer #2: Yes

4. Have the authors made all data underlying the findings in their manuscript fully available?

Reviewer #1: Yes

Reviewer #2: Yes

5. Is the manuscript presented in an intelligible fashion and written in standard English?

Reviewer #1: Yes

Reviewer #2: Yes

6. Review Comments to the Author

Reviewer #1: I'm satisfied with the answers to my comments, reason why I consider that it is possible to be admitted

Reviewer #2: The authors answered to all the raised questions and the impression is that the quality of the manuscript has been overall improved.

Some minor concerns and one question to the authors:

1) Discussion: page 20, lines 1-4, the sentence “However… sex-related differences in LVM” appears wordy and disconnected, please reprhase.

2) Discussion: page 20, lines 7-10. The sentence “But…. impedance analysis” also appears wordy. Please check the punctuation to make the sentence easier to read.

3) This analysis has been performed on a sportive pediatric population, though not professional athletes have been considered. Do you think that your results should be applied to all pediatric populations regarding the participation in sport? Or do you think that the different kind and amount of sport performed by the two different sexes might have in some manner influenced the results? This question is raised while evaluating a young patient with a suspicion of cardiomyopathy, especially in they “grey” zone. Should we correct the LVM for sex even in this population? The authors should address this topic in the discussion to give further clinical impact to your work.

7. PLOS authors have the option to publish the peer review history of their article (what does this mean?). If published, this will include your full peer review and any attached files.

Reviewer #1: Yes: Marc Abulí Lluch

Reviewer #2: No

---

## [Author Response · Author response to Decision Letter 1]

6 Jun 2020

Response to reviewers

Reviewer #2: The authors answered to all the raised questions and the impression is that the quality of the manuscript has been overall improved.

Some minor concerns and one question to the authors:

1) Discussion: page 20, lines 1-4, the sentence “However… sex-related differences in LVM” appears wordy and disconnected, please reprhase.

Thank you very much for the comment. As you suggested, we've rephrased this sentence as follow:

“According to some researchers, there is no difference between boys and girls in the course of changes of LVM against LBM. They argue that since lean body mass (or fat-free mass, FFM) is the strongest determinant of LVM, the sex-related differences in LVM can be explained by the differences in LBM between boys and girls.”

2) Discussion: page 20, lines 7-10. The sentence “But…. impedance analysis” also appears wordy. Please check the punctuation to make the sentence easier to read.

Thank you very much for the comment. As you suggested, we've rephrased this sentence as follow:

“Since LBM cannot be measured directly, advanced indirect methods such as double X-ray absorptiometry, computed tomography, magnetic resonance imaging, or bioelectrical impedance analysis are required for reliable measurements.”

3) This analysis has been performed on a sportive pediatric population, though not professional athletes have been considered. Do you think that your results should be applied to all pediatric populations regarding the participation in sport? Or do you think that the different kind and amount of sport performed by the two different sexes might have in some manner influenced the results? This question is raised while evaluating a young patient with a suspicion of cardiomyopathy, especially in they “grey” zone. Should we correct the LVM for sex even in this population? The authors should address this topic in the discussion to give further clinical impact to your work.

The children and adolescents whose echocardiographic data were used in this study were athletes practicing mostly amateur-level sport. An athletic population is considered as the special one because regular exercise contributes to increasing cardiac size, which is an adaptive response causing increased exercise capacity. As the training volume and intensity have to be adapted to the athlete exercise capacity, the absolute training workloads in girls are lower than in boys. Therefore, we cannot exclude that the different intensity of training in girls and boys additionally contributed to the fact that for a given body size LVM in boys was higher than in girls.

Thus, we agree that the specificity of the population examined in this study may have "in some manner influenced the results," but this is not a limitation; it is an advantage. The group selection allowed us to demonstrate that there are differences in cardiac size between boys and girls, which cannot be explained by differences in body size.

At the same time, we know that regardless of whether they practice sports or not, there are differences in exercise capacity between boys and girls, which cannot be explained by differences in body size. In general, the exercise capacity in boys is greater than in girls. Perhaps in the youngest, the differences are nonsignificant, but with age, the gap is growing. Athletic training might only augment the difference in exercise capacity between boys and girls within the population. The same goes for cardiac size. 

Therefore, we are convinced that for all pediatric populations, one should use LVM normative data that were developed separately for girls and boys.

We have introduced relevant information to the discussion and the study limitations sections.

---

## [Decision Letter · Decision Letter 2]

22 Jun 2020

PONE-D-20-05310R2

Left ventricular mass normalization in child and adolescent athletes must account for sex differences.

PLOS ONE

Dear Dr. Krysztofiak,

Thank you for submitting your manuscript to PLOS ONE. After careful consideration, we feel that it has merit but does not fully meet PLOS ONE’s publication criteria as it currently stands. Therefore, we invite you to submit a revised version of the manuscript that addresses the points raised during the review process.

Your manuscript was sent back to the previous Reviewers and a few minor points should be addressed.

We look forward to receiving your revised manuscript.

Kind regards,

Daniel M. Johnson, PhD

Academic Editor

PLOS ONE

Reviewers' comments:

Reviewer's Responses to Questions

**Comments to the Author**

1. If the authors have adequately addressed your comments raised in a previous round of review and you feel that this manuscript is now acceptable for publication, you may indicate that here to bypass the “Comments to the Author” section, enter your conflict of interest statement in the “Confidential to Editor” section, and submit your "Accept" recommendation.

Reviewer #2: (No Response)

2. Is the manuscript technically sound, and do the data support the conclusions?

Reviewer #2: Yes

3. Has the statistical analysis been performed appropriately and rigorously? 

Reviewer #2: Yes

4. Have the authors made all data underlying the findings in their manuscript fully available?

Reviewer #2: Yes

5. Is the manuscript presented in an intelligible fashion and written in standard English?

Reviewer #2: No

6. Review Comments to the Author

Reviewer #2: The authors have adequately rephrased the sentences as suggested (point 1 and 2).

However, the new section added to the discussion (point 3 – “Since the study has been performed … for cardiac size.”) needs rephrasing and implementation; specifically, a) informal registry (“..we are convinced that yes, … the same goes for …”), b) incorrect tenses or grammar errors (“..as the special one because regular exercise contribute to …. the gap is growing.”), c) be more precise when stating that in the youngest differences in exercise capacity are non-significant, from which age should we consider that gap significant? Please, clarify the relationship between exercise capacity and LVM in non-athletes and how this information impacts on your issue.

7. PLOS authors have the option to publish the peer review history of their article (what does this mean?). If published, this will include your full peer review and any attached files.

Reviewer #2: No

---

## [Author Response · Author response to Decision Letter 2]

5 Jul 2020

Response to reviewer

Reviewer #2: 

The authors have adequately rephrased the sentences as suggested (point 1 and 2).

However, the new section added to the discussion (point 3 – "Since the study has been performed … for cardiac size.") needs rephrasing and implementation; specifically, a) informal registry ("..we are convinced that yes, … the same goes for …"), b) incorrect tenses or grammar errors ("..as the special one because regular exercise contribute to …. the gap is growing."), c) be more precise when stating that in the youngest differences in exercise capacity are non-significant, from which age should we consider that gap significant? Please, clarify the relationship between exercise capacity and LVM in non-athletes and how this information impacts on your issue.

Thank you, we appreciate your comments. The errors you pointed out should not happen. We have rewritten this section and introduced additional information addressing substantive issues you raised. 

This section now reads as follows:

"The study participants were child and adolescent athletes. The athletic population may be considered special because regular exercise contributes to an increase in cardiac size [15], and specific LVM normative data are recommended for child and adolescent athletes [20]. A question may thus arise as to whether the results of the study should be applied to all children and adolescents.

The increase in cardiac size in response to exercise is an adaptive phenomenon linked to the improvement of exercise capacity. Not only athletes but generally all healthy children, both boys, and girls, have higher LVM when their exercise capacity is higher [50]. However, boys have higher exercise capacity than girls of the same age, and this is true for both athletic and non-athletic populations [51]. Perhaps in the youngest, the difference is not seen, but after the age of about twelve, it becomes significant [52]. The same pattern is observed in the case of LVM. Before puberty, LVM in healthy boys and girls is similar, and at puberty, the difference between boys and girls becomes evident, with higher LVM in boys [53].

Thus, higher exercise capacity is associated with higher LVM, and within comparable groups of healthy children, especially adolescents, boys have higher exercise capacity and LVM than girls. That is not solely specific to athletes. Normative data for LVM should be developed separately for boys and girls for all children and adolescents, regardless of whether they practice sports or not."

---

## [Editor Report · Decision Letter 3]

13 Jul 2020

Left ventricular mass normalization in child and adolescent athletes must account for sex differences.

PONE-D-20-05310R3

Dear Dr. Krysztofiak,

We’re pleased to inform you that your manuscript has been judged scientifically suitable for publication and will be formally accepted for publication once it meets all outstanding technical requirements.

Kind regards,

Daniel M. Johnson, PhD

Academic Editor

PLOS ONE
---

## [Editor Report · Acceptance letter]

15 Jul 2020

PONE-D-20-05310R3 

Left ventricular mass normalization in child and adolescent athletes must account for sex differences. 

Dear Dr. Krysztofiak:

I'm pleased to inform you that your manuscript has been deemed suitable for publication in PLOS ONE. Congratulations! Your manuscript is now with our production department. 

Kind regards, 

on behalf of

Dr. Daniel M. Johnson 

Academic Editor

PLOS ONE